# Clinical Characteristics, Treatment Modalities, and Potential Contributing and Prognostic Factors in Patients with Bone Metastases from Gynecological Cancers: A Systematic Review

**DOI:** 10.3390/diagnostics11091626

**Published:** 2021-09-06

**Authors:** Francesca Salamanna, Anna Myriam Perrone, Deyanira Contartese, Veronica Borsari, Alessandro Gasbarrini, Silvia Terzi, Pierandrea De Iaco, Milena Fini

**Affiliations:** 1Complex Structure Surgical Sciences and Technologies, IRCCS Istituto Ortopedico Rizzoli, 40136 Bologna, Italy; francesca.salamanna@ior.it (F.S.); veronica.borsari@ior.it (V.B.); milena.fini@ior.it (M.F.); 2Division of Oncologic Gynecology, IRCCS Azienda Ospedaliero-Universitaria di Bologna, 40138 Bologna, Italy; myriam.perrone@aosp.bo.it (A.M.P.); pierandrea.deiaco@unibo.it (P.D.I.); 3Department of Medical and Surgical Sciences (DIMEC), University of Bologna, 40126 Bologna, Italy; 4Centro di Studio e Ricerca delle Neoplasie Ginecologiche (CSR), University of Bologna, 40138 Bologna, Italy; 5Department of Oncologic and Degenerative Spine Surgery, IRCCS Istituto Ortopedico Rizzoli, 40136 Bologna, Italy; alessandro.gasbarrini@ior.it (A.G.); silvia.terzi@ior.it (S.T.)

**Keywords:** bone metastases, gynecological cancer, clinical studies, systematic review

## Abstract

The purpose of this study is to review the clinical characteristics, treatment modalities, and potential contributing and prognostic factors of bone metastases from gynecological cancers (GCs). A systematic literature search on PubMed, Scopus, Web of Science Core Collection and Cochrane Central Register of Controlled Trials databases was conducted. Thirty-one studies, all retrospective, were included in this review, for a total of 2880 patients with GC bone metastases. Primary tumors leading to bone metastases included endometrial cancer (EC), cervical cancer (CC), ovarian cancer (OC), uterine sarcoma (US) and vulvar cancer (VuC), mainly with an International Federation of Gynecology and Obstetrics (FIGO) Stage of III and IV. The main bone metastatic lesion site was the vertebral column, followed by the pelvic bone and lower extremity bones. The median survival rate after bone metastases diagnosis ranged from 3.0 to 45 months. The most frequent treatments were palliative and included radiotherapy and chemotherapy, followed by surgery. The findings of this review give a first dataset for a greater understanding of GC bone metastases that could help clinicians move toward a more “personalized” and thus more effective patient management.

## 1. Introduction

Bone is a common site of metastases and frequently indicates a short-term prognosis in patients with cancer. The acute effects of bone metastases include skeletal-related events (SREs; characterized by pathological fracture, the need for radiotherapy or bone surgery, spinal cord compression, and hypercalcemia), which may compromise patients’ physiological function and quality of life. Once cancer spreads to the bones, it can seldom be cured, but frequently it can be treated to reduce its growth. Breast, prostate, lung, kidney, and thyroid cancers are most likely to spread to the bone, and numerous studies in have analyzed their clinical characteristics, treatment, and potential contributing and prognostic factors [1]. However, other cancer types can also metastasize to bone; the increase in overall survival in patients with primary and/or metastatic cancer increases the probability that patients can develop bone metastases during their disease. This is also true for gynecological cancers (GCs), which include cervical, ovarian, uterine, vaginal, and vulvar malignancies. Worldwide, GCs affect millions of women across all ages [2]. It is estimated that in the United States, the number of women with GCs is approximately 80,000/year [3]. However, differences in the incidence of GCs in diverse areas of the world have been detected. In the European Union, the estimated number of newly diagnosed cases is higher, with about 200,000 women/year [4]. Despite the enormous progress and advancements in the preventive, diagnostic, and therapeutic interventions for GCs, distant metastases and recurrence are still the leading cause of death in patients. Metastases from GCs differ depending on the cancer type. Cervical cancer (CC), endometrial cancer (EC), and uterine sarcomas (US) mainly spread to the lung and liver, while ovarian cancer (OC) primarily metastasizes locally within the peritoneum and pelvis [5,6,7]. However, distant sites such as bone may be also involved [8,9,10,11,12]. Bone metastases from GCs were detected in about 1.1–5.2% of patients with CC [8], in ~1.2% of patients with OC [9], in 2–8% of patients with EC [10], and rarely in vulvar (VuC) [11] and vaginal cancers (VaC) [12]. Despite the uncommon incidence of GC bone metastases, the prolonged local disease control and the easy availability of advanced imaging techniques are increasing this incidence, and more and more anecdotal evidence is reported in the literature [8,13,14]. Not unexpectedly, these complications seriously affect quality of life, performance status, independent functioning, and survival. Conventionally, radiotherapy is considered the first line treatment in the management of GC metastatic bone lesions; however, impending or existing pathological fractures, spinal cord compression, unbearable pain, and resistance to radiotherapy are indications/complications that in most cases require a surgical approach [15]. Obviously, considering GC heterogeneity, i.e., entity, type, stage, grade, as well as patient heterogeneity (i.e., physical status, age, life styles, menopausal status), the “optimal” management can change [15]. Consequently, comparability between patients with GC bone metastasis in clinical studies is still limited and complex due to specific differences between tumors of the same type in different patients and between cancer cells within a tumor. This complexity is even more accentuated since few single-center, multi-center, and population-based studies report specific details on patients with GC bone metastasis. Furthermore, a systematic review of data on different types of GC bone metastasis, also considering all the anecdotal evidence present in literature, is, to date, lacking. Thus, a general overview of indications on the evaluation, incidence, complications, prognostic characteristics, and management of GCs that metastasize to bone remains a challenge for both gynecologic oncologists and researchers. Here, we conducted a systematic review, focusing on clinical characteristics, treatment, and potential contributing and prognostic factors of bone metastases in GC-affected patients in order to have a first dataset able to give a greater understanding of this pathological condition and to guide clinicians towards a more “personalized” patient management.

## 2. Results

### 2.1. Study Selection and Characteristics

The initial literature search retrieved 4102 studies. Of those, 645 studies were identified using PubMed, 2469 using Scopus, 939 in the Web of Science Core Collection, and 49 using the Cochrane Central Register of Controlled Trials. After screening the title and abstract, 262 articles were submitted to a public reference manager to eliminate duplicate articles. The resulting 79 complete articles were then reviewed to establish whether the publications met the inclusion criteria, and 31 studies were considered eligible for this review. All the included studies were retrospective. Search strategy and study inclusion and exclusion criteria are detailed in Figure 1.

### 2.2. Assessment of Methodological Quality

The risks of bias assessments for each study are summarized in Table 1. Of the 31 articles eligible for the review, we found that all studies were retrospective. Using the NIH Quality Assessment Tool for Observational Cohort Studies [16,17], we rated one retrospective study at a “good” quality rating and 30 studies at a “fair” quality rating (Table 1). For the 30 retrospective studies at a “fair” quality rating, the principal missing quality assessment criteria were sample size justification and blinded assessors to the exposure of participants (Table 1). However, 14 studies also showed bias for the description of the study population and for the presence of potential confounding variables not measured and/or statistically adjusted (Table 1).

### 2.3. Characteristics of the Studies

Characteristics of the studies regarding geographical region, execution date, type, patient number, bone metastases occurrence, median/mean age, menopausal status, primary tumor histopathology and clinical stage and grade, survival, and/or follow-up period after bone metastases are described in Table 2. The 31 studies included in this review, selected according to the inclusion criteria, were all published between 2013 and 2021, but the time horizon covered for patient selection ranged between 1984 and 2019. All the studies were retrospective; 4 were on OC, 11 on EC, 2 on US, 1 on VuC, and 13 on CC. No studies on VaC bone metastases were founded with our search strategy.

For OC, we selected four retrospectives studies. In these studies, the median survival times after bone metastases diagnosis ranged from 3.0 to 21.5 months [20]. A study by Zhang et al. also reported the mean survival time considering a cohort of patients without GC bone metastases treatment (3.0 months) and a cohort of patients treated with chemotherapy + radiotherapy + radionuclide therapy (21.5 months) [20]. The number of patients enrolled in the studies ranged from 1481 [18] to 32,178 [21], with a bone metastases occurrence that ranged from 1.0% [21] to 2.1% [18]. The median patient age at the time of diagnosis of bone metastases ranged from 46.7 [20] to 65.6 [21]. Premenopausal status was reported only in one study, which included 42.3% premenopausal women [19]. At the time of OC diagnosis, most patients had the International Federation of Gynecology and Obstetrics (FIGO) Stage III/IV with Grade II/III [18,19,20,21].

For EC, we founded 11 retrospectives studies. In these studies, the median survival time after bone metastases diagnosis ranged from 4.0 to 18.0 months [24,25], with a study that underlined the difference in median survival between patients with Stage III (4 months) and patients with Stage I (17.5 months). The bone metastases occurrence in this cohort of patients ranged from 0.5% [27] to 15.4% [22]; this difference could be due to the different number of enrolled patients, which ranged from 403 [23] to 69,027 in the case of the lowest occurrence [27]. The median patient age at the time of diagnosis of bone metastases ranged from 47.5 [26] to 75.5 [29]. A postmenopausal status was reported in 5/12 studies, while the others did not specify the menopausal status [23,24,29,30,31]. At the time of EC diagnosis, most patients had a FIGO Stage III–IV with Grade II–III.

For US, we found two retrospectives studies, with a median survival time after bone metastases diagnosis that ranged from 22.0 to 45.0 months [33,34]. Both retrospective studies reported cohort of patients with uterine leiomyosarcoma, with a bone metastases occurrence that ranged from 13.8% [33] to 32.7% [34]; the number of enrolled patients varied from 113 [34] to 130 [33]. The median patient age at the time of diagnosis of bone metastases was ~53 years for both retrospective studies. The postmenopausal status was not specified. At the time of US diagnosis, the FIGO Stage ranged from I to IV.

VuC bone metastases were detailed in one retrospective study, which described a cohort of 391 patients with an occurrence of bone metastases of 1.2% [35]. The patients’ median age was 60 years, and the study included pre- and postmenopausal women with FIGO Stage I and Grade II–III. All the recruited patients had a squamous-cell VuC with an overall survival after bone metastasis of 36 months.

CC bone metastases represented ~42% of all studies found with our research strategy, with 13 retrospective studies. These studies showed a median survival time after bone metastases diagnosis that ranged from 7.0 to 34.0 months [38,42]. One study specified an overall survival of 14 months for patients with bone metastases only and a lower overall survival, of 5 months, for patients with extra-osseous metastases [39]. Additionally, Nartthanarung et al. described differences in the overall survival considering the patients mean age, with an overall survival of 21 months in patients < 45 years and an overall survival of 34 months for patients > 45 years [42]. Almost all the retrospective studies reported a cohort of patients with squamous-cell cervical cancer, with a bone metastases occurrence that ranged from 1% [38] to 35.9% [47] and with cohorts that varied from 99 [44] to 19,363 patients [46]. The median patient age at the time of diagnosis of bone metastases ranged from 40 to 65 years [48]. Most studies included both pre- and postmenopausal women, with a FIGO Stage that ranged from I to IV as well as a Grade that ranged from I to III. 

### 2.4. Bone Metastases Diagnosis and Main Characteristics

Bone metastases diagnosis methodology, the length of the bone metastases-free interval, the number and sites of bone metastases, the bone metastases symptoms, and the presence of extraosseous metastases were extracted and are reported in Table 3.

For OC bone metastases, the most common imaging modalities for diagnosing were computed tomography (CT), magnetic resonance imaging (MRI), radiography (X-ray) and, for one study, bone scintigraphy (BS). The bone metastases-free interval ranged from 5 years [20] to 1 year [19,20], in relation to the FIGO stage. In some patients the bone metastases diagnosis was also made at time of the OC primary diagnosis [19,20]. In this context, Zhang et al. reported bone metastases diagnosis at the same time of OC primary diagnosis prevalently for patients with FIGO Stage IV [20]. The most reported bone metastases site was the spine [19,20], followed by the pelvic bone, ribs, sternum, bones of the extremities, and skull. Bone metastases were single or multiple, with pain as the main reported symptom [19,20]. Extraosseous metastases were frequently present, with the lung, liver, and brain as the main reported organs [18,20].

EC bone metastases were mostly diagnosed with PET/CT or CT alone, followed by MRI and X-ray [22,23,24,25,26,27,28,29,30,31,32]. The bone metastases-free interval ranged from 6 months [23] to 19.5 months [31]. In several patients with EC, the bone metastases diagnosis was made at the time of the primary cancer diagnosis [28,31,32]. Bone metastases sites were both single and multiple, with spine, pelvis, tibia, and femur as the most reported anatomical sites of bone metastases and pain as the most reported bone metastases symptom [28,30,31,32]. Extraosseous metastases were commonly present, but none of the examined studies specified the site [22,23,24,25,26,27,28,29,30,31,32].

In the studies on US, the bone metastases were diagnosed by CT, showing a bone metastasis-free interval that ranged from 7 months [34] to 3 years [33]. Bone metastases sites were single in one study [33] and both single and multiple in the other study [34], with the presence of extraosseous metastases [33,34].

In the only study on VuC, the bone metastases were diagnosed with CT and X-ray and presented a bone metastasis-free interval of 13 months [35]. Bone metastases were in single and multiple sites and metastasized to the spine, pelvis, and thigh [35]. Extraosseous metastases were reported in the lungs and liver [35].

For CC bone metastases studies, the most common imaging modalities for diagnosing were CT and MRI, followed by PET/CT, FDG-PET/CT, X-ray, and BS. The bone metastases-free interval ranged from 10 months [36] to 27 months [45]. However, in most patients, the bone metastases diagnosis was made at time of the primary tumor diagnosis [39,40,46]. The main reported bone metastases sites were the spine and pelvis, with bone metastases present in single and multiple sites and with pain as the principal reported symptom [36,37,39,40,41,42,45]. Extraosseous metastases were described by almost all studies, but none of them specified the anatomical sites.

### 2.5. Complication of Bone Metastases and Development of Skeletal-Related Events

Patients with GCs that develop bone metastases prevalently have osteolytic lesions that can lead to complications, referred to as SREs. At present, in these cases the most frequent treatment is to relieve symptoms. The purpose of treatment is to reduce pain, prevent pathological fracture occurrence, inhibit the disease progression, improve function and quality of life, and prolong survival time. Treatment options include comprehensive anti-tumor therapy, i.e., surgery, radiotherapy, chemotherapy, and in some cases bone resorption suppression therapy and other palliative treatments. Complications from bone metastases, i.e., SRE percentage, bone pain, pathological fractures, spinal cord compression, and hypercalcemia, as well as first-line therapies for patients with GC bone metastases are summarized in Table 4.

For OC bone metastases, only 1/4 retrospective studies [19] reported the SRE percentage (33%). Bone pain was reported in the majority of the studies (*n* = 3/4), while pathological fractures were detected in two studies [19,20]. Spinal cord compression was reported only in one study [20], while it was not present in two studies [19,21] and not specified in one study [18]. The absence of hypercalcemia was specified in one study [20], while the other studies did not specify the calcium level. First-line treatments for patients with bone metastases from OC were chemotherapy, radiotherapy, and surgery.

The percentage of SREs for EC bone metastases was reported only in the study by McEachron et al. [28] and was 1%. Bone pain was reported and specified in three studies (3/11), while pathological fractures were reported in 3/11 studies. No studies reported and specified the presence/absence of spinal cord compression as well as data on hypercalcemia. First-line treatments for patients with EC bone metastases were specified in most of the analyzed studies (9/11) and generally were radiotherapy and chemotherapy, followed by surgery.

For US and VuC bone metastases, data on SREs and the development of bone metastases complications were seldom reported [33,34,35]. None of the studies on these GC bone metastases reported data on SRE%, calcium levels, fractures and spinal bone compression, and pathological fractures. First-line treatments for these GC bone metastases were radiotherapy, chemotherapy, and surgery.

None of the 13 retrospective studies on CC bone metastases reported the SRE%, potential presence of pathological fractures, and spinal cord compression, while two of them reported on hypercalcemia [40,45] and four reported on bone pain [36,40,42,45]. First-line treatments for CC bone metastases were principally radiotherapy and chemotherapy, followed by surgery and other palliative treatments.

## 3. Discussion

Although over the years numerous studies have improved our knowledge on bone metastases secondary to GCs, to date, important gaps are still present due to few relevant reports and a paucity of clinical data. Specifically, there is limited information on clinical datasets, patients’ specific characteristics, prognostic and predictive factors among patients, metastatic tumors, and treatments. Additionally, the heterogeneity of GC types/subtypes, their histopathology, and the variability across geographical regions (including screening method and missing, underreported, or incorrect data) further complicates the understanding of certain clinical features and specific therapeutic approaches to GC bone metastases. Thus, to better highlight and clarify these aspects, we carried out a review to systematically and qualitatively describe and analyze the clinical characteristics, treatment, and potential contributing and potential prognostic factors of GC bone metastases.

The incidence of GC bone metastases remained at about 1.1–5.2% for CC, 2–8% for EC, 1.2% for OC, and <1% for other GCs over decades. However, current methods for primary disease control and the easy accessibility to advanced imaging techniques have increased GC bone metastases incidence in the last 10 years. In our review, the occurrence of GC bone metastases was 1.0–35.9% for CC, 0.5–15.4% for EC, 1.0–2.1% for OC, 13.8–32.7% for US, and 1.2% for VuC. Although the lytic component was predominant in these GCs, both processes are usually accelerated within the bone metastasis, resulting in “mixed” lesions in which both lytic and sclerotic components are visible. In fact, autopsy studies revealed that bone metastases can be heterogeneous within a single patient, that is, osteolytic at one site and osteoblastic or mixed at another site [49,50]. This review underlined that GC bone metastases incidence was higher in patients with advanced stage disease (III/IV), though also present in a lower percentage of patients with low tumor stage (I/II) and grade (I). Diagnosis was made mostly via CT scans and X-rays. These methods of diagnosis assess the stromal reaction to the presence of cancer cells within the bone marrow rather than depicting the cancer foci themselves. This lack of direct depiction of tumor foci limits early metastatic detection and assessment of the response of bone metastases to treatment. In fact, to limit this problem and to improve the assessment of GC metastatic bone disease, several studies used high-sensitivity imaging methods such as a PET scan, with various radiotracers, and whole-body MRIs [51]. Once bone metastasis has been identified, the treatment is frequently multimodal and interdisciplinary. In this review, the treatment modalities most often used for GC bone metastases were chemotherapy and radiotherapy. The treatment of general pain was the common factor among all GC types, but instability, bone fractures, and spinal cord compression were also present. In these cases, bisphosphonates (which are characterized by ease of administration, long duration of action, safety, and effectiveness) such as zoledronic acid, alendronate, risedronate, and the third-generation agent ibandronate, or RANKL inhibitor, i.e., denosumab, were also used to reduce the incidence of fractures or spinal cord compressions and to relieve diffuse pain [52]. In addition, to relieve suffering and provide the best possible quality of life for both patients and their families, no matter what the treatment course, palliative care (i.e., nutrition, rehabilitation, control of symptoms such as pain, dyspnea, fear, and anxiety) was also used in addition to chemotherapy and radiotherapy [53,54]. Good palliative care preserves the patient’s quality of life through a predominantly multidimensional approach to symptom control of evolving multi-morbidity and side effects of the primary treatment. In most studies, surgical intervention was also carried out, mostly to manage the structural complications associated with bone destruction and/or nerve compression. However, despite these interventions, the patient’s overall survival after bone metastasis diagnosis and treatments was poor and coincided with previous research findings. The longest median overall survival time found in this review was 45 months; however, most studies indicate a median overall survival of ~12 months. In agreement with the literature, our review revealed that GC bone metastases most affected the axial skeleton and in particular the vertebral column [13]. The causes of this trend are not yet fully understood, but it has been hypothesized that the cellular and molecular characteristics of cancer cells and the tissues to which they metastasize are critical and affect the pattern of metastatic spread; however contradictory or inconclusive results have been reported [14,55]. As shown by this review, these conflicting results can likely be explained by the high heterogeneity in patient populations, cancer treatment, and study methodology. Interestingly, our review showed that most patients with GC bone metastases were in menopause; thus, it is possible to speculate that menopausal status can have an influence on the development of GC bone metastases, confirming the protective influence of estrogen on bone density. This aspect was also confirmed for breast cancer bone metastases, in which bimodal interactions between pre-existing estrogen deficiency due to osteoporosis (and related factors) and bone metastasis development was detected. Looking at the literature, it is clear that bone metastases and estrogen deficiency present numerous sharing factors, e.g., disorders in monocyte and macrophage functions and consequent alterations in immune functions, alterations in the balance between pro- and anti-inflammatory regulators, improvement in angiogenesis, platelet deregulation, thromboembolism events, extracellular matrix components, and hormone changes [14]. However, several other contributing factors, such as specific cellular and molecular alterations, gene signature changes, and specific relationships in bone remodeling and primary tumor cells, could further influence the development of GC bone metastases, also considering that bone colonization by malignant cells includes complex interplays between tumor cells and resident bone cells (i.e., osteoclasts, osteoblasts, and osteocytes), bone marrow cells, and the bone matrix. These aspects also suggest that effective treatment strategies for bone metastasis should consider an association of bone-targeted agents in combination with different local and/or systemic anti-tumor strategies for the primary tumor. Thus, a multidisciplinary approach involving oncologists, radiotherapists, orthopedic surgeons, intervention radiologists, pain specialists, and palliative care physicians is mandatory. Furthermore, other specific drawbacks of existing clinical treatments, e.g., systemic administration of antiresorptive drugs and/or anti-tumor agents and/or radiopharmaceuticals, involve adverse effects of these treatments on normal bone metabolism, which may result in harmful outcomes for treated patients. Thus, new and advanced therapeutic options are needed. In this context, emerging bone-targeting therapies are based on the use of anabolic agents or molecules with a dual action, i.e., therapies able to activate osteoblasts and inhibit osteoclasts, or on specific drugs able to target the bone microenvironment. Other promising targets comprise molecules such as small non-coding RNAs (miRNAs), which function as key regulators of various biological and pathological processes, including physiological bone remodeling and bone metastasis. Several studies evaluated the efficacy of these miRNAs as antimetastatic agents and/or as predictors of metastatic bone disease. In detail, recent studies provided evidence that miR-338-3p, miR-208a-5p, miR-4443, and miR-5195-3p contribute to metastasis and GC tumorigenesis via multiple mechanisms [56,57,58]. Furthermore, several authors also evaluated circulating tumor DNA (ctDNA) and circulating tumor cells (CTCs) as alternative “liquid biopsies” modalities in patients with GC metastases [59,60,61]. However, the use of cell-free DNA (cfDNA) could be associated with the prediction of response to targeted therapy and the detection of subpopulations/gene signatures of cfDNA, allowing diagnosis, stratification of therapy, and more accurate prognosis. Despite this wide platform of strategies and approaches, their use in the clinical setting remains limited since their application would require a multistage procedure with multiannual and toilsome research approaches. In this context, the development of specific algorithms through recent technological advances related to artificial intelligence could offer a potential alternative and contribution to analyze panels of specific markers/parameters associated with imaging data to determine individual risk factors for GC bone metastases, thus triggering the use of therapies/treatments able to increase the bone-disease-free interval and the survival time after bone metastases.

Our review has several limitations. First, the retrospective nature of the analyzed studies makes them subject to selection bias. Second, most of the included studies had a limited sample size (only 8/31 studies included had more than 100 patients with GC bone metastases), thus lacking sufficient statistical power. Third, there was a marked heterogeneity in patient populations. And finally, methods for detecting and characterizing GC bone metastases complications and skeletal-related events are often not reported.

In conclusion, the results of this systematic review give the first dataset for a greater understanding of GC bone metastases that could be able to guide clinicians towards a more “personalized” patient management. Obviously, further well-designed clinical studies able to accurately determine the risk of developing bone metastases after a diagnosis of GCs are mandatory. We look forward to future prospective and large population-based research on this complex issue, which we hope will improve and increase quality of life and patient survival time.

## 4. Materials and Methods

### 4.1. Eligibility Criteria

The PICOS framework (population, intervention, comparison, outcomes, study design) was used to formulate the questions for this study: (1) patients with bone metastases from GCs, i.e., OC, EC, US, VuC, VaC, CC (Population); (2) not applicable (Interventions); (3) not applicable (Comparisons); (4) studies that reported incidence, complication (e.g., skeletal-related events), and prognostic characteristics (age, primary tumor aggressiveness, e.g., subtype, stage, grade, etc.) of bone metastases in patients with GCs (Outcomes); and (4) cohort studies, including randomized controlled trials (Study Design). The focused question was “What are the clinical characteristics, treatment, and potential contributing and prognostic factors of bone metastases from GCs in the clinical setting?”. Studies from 24 May 2011 to 19 August 2021 were included in this review if they met the PICOS criteria.

We excluded studies investigating (1) only primary GCs, (2) metastatic sites different than bone, (3) cancers other than bone metastatic GCs, (4) different types of GC bone metastases, considered all together. Additionally, we excluded case reports, case series, abstracts, editorials, letters, comments to the editor, reviews, meta-analyses, book chapters, and articles not written in English.

### 4.2. Information Source and Search Strategies

Our literature review involved a systematic search conducted on 19 August 2021. We performed our review according to the PRISMA 2020 statement [62]. The search was carried out on PubMed, Scopus, Web of Science Core Collection, and Cochrane Central Register of Controlled Trials databases to identify studies on bone metastases from GCs, specifically OC, EC, US, VuC, VaC, and CC. A search was conducted combining the terms “bone metastasis” AND “ovarian cancer”, “bone metastasis” AND “endometrial cancer”, “bone metastasis” AND “uterine sarcoma”, “bone metastasis” AND “vulvar cancer”, “bone metastasis” AND “vaginal cancer”, “bone metastasis” AND “cervical cancer”; for each of these terms, free words and controlled vocabulary specific to each bibliographic database were combined using the operator “OR”. The combination of free vocabulary and/or Medical Subject Heading (MeSH) terms for the identification of studies in PubMed, Scopus, Web of Science Core Collection, and Cochrane Central Register of Controlled Trials are reported in Table 5.

### 4.3. Study Selection and Data Extraction

Possible relevant articles were screened, using title and abstract, by two reviewers, and articles that did not meet the inclusion criteria were excluded. After screening the title and abstract, articles were submitted to a public reference manager to eliminate duplicates. Subsequently, the remaining full-text articles were retrieved and examined by two reviewers. Any disagreement was resolved through discussion until a consensus was reached or with the involvement of a third reviewer.

Data from the retrieved studies were tabulated taking into consideration the studies’ general characteristics, the GC bone metastases diagnosis and main characteristics, and the development of complications, e.g., skeletal-related events, in GCs bone metastases.

### 4.4. Assessment of Methodological Quality

Two reviewers independently assessed the methodological quality of selected studies. In case of disagreement, they attempted to reach consensus; if this failed, a third reviewer made the final decision. The methodological quality of cohort studies was assessed using the Quality Assessment Tools of the National Heart, Lung, and Blood Institute (NHLBI) [16,17].

## Figures and Tables

**Figure 1 diagnostics-11-01626-f001:**
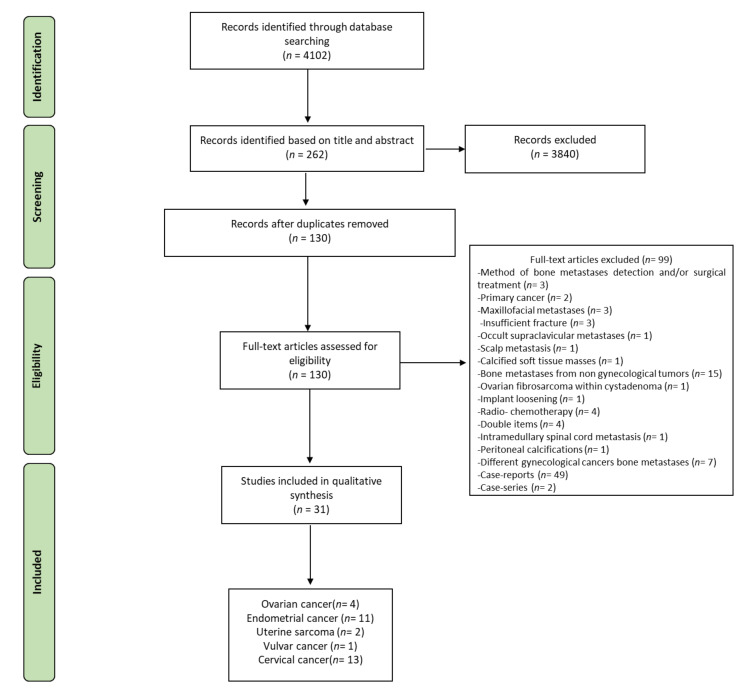
Preferred Reporting Items for Systematic Reviews and Meta-Analyses (PRISMA) flowchart for the selection of studies.

**Table 1 diagnostics-11-01626-t001:** Quality assessment tool for retrospective cohort studies.

Reference	Criteria													
	1	2	3	4	5	6	7	8	9	10	11	12	13	14
Ovarian cancer														
Deng et al. 2018 [18]							NA	NA		NA				
Sehouli et al. 2013 [19]							NA	NA		NA				
Zhang et al. 2013 [20]					CD		NA	NA		NA				
Zhang et al. 2019 [21]							NA	NA		NA				
Endometrial cancer														
Guo et al. 2020 [22]							NA	NA		NA				
Hong et al. 2021 [23]							NA	NA		NA				
Kimyon et al. 2016 [24]							NA	NA		NA				
Li et al. 2020 [25]							NA	NA		NA				
Liu et al. 2020 [26]							NA	NA		NA				
Mao et al. 2020 [27]							NA	NA		NA				
McEachron et al. 2020 [28]							NA	NA		NA				
Ouldamer et al. 2019 [29]							NA	NA		NA				
Takeshita et al. 2016 [30]							NA	NA		NA				
Uccella et al. 2013 [31]							NA	NA		NA				
Yoon et al. 2014 [32]							NA	NA		NA				
Uterine sarcoma														
Bartosh et al. 2017 [33]							NA	NA		NA				
Tirumani et al. 2014 [34]							NA	NA		NA				
Vulvar cancer														
Prieske et al. 2016 [35]							NA	NA		NA				
Cervical cancer														
Kanayama et al. 2015 [36]							NA	NA		NA				
Kocaer et al. 2018 [37]							NA	NA		NA				
Lin et al. 2018 [38]							NA	NA		NA				
Makino et al. 2016 [39]							NA	NA		NA				
Manders et al. 2018 [40]							NA	NA		NA				
Matsumiya et al. 2016 [41]							NA	NA		NA				
Nartthanarung et al. 2014 [42]							NA	NA		NA				
Sethi et al. 2019 [43]							NA	NA		NA				
Yin et al. 2019 [44]							NA	NA		NA				
Yoon et al. 2013 [45]							NA	NA		NA				
Zhang et al. 2018 [46]							NA	NA		NA				
Zhang et al. 2020 [47]							NA	NA		NA				
Zhou et al. 2020 [48]							NA	NA		NA				

**1.** Was the research question or objective in this paper clearly stated? **2.** Was the study population clearly specified and defined? **3.** Was the participation rate of eligible persons at least 50%? **4.** Were all the subjects selected or recruited from the same or similar populations (including the same time period)? Were inclusion and exclusion criteria for being in the study prespecified and applied uniformly to all participants? **5.** Was a sample size justification, power description, or variance and effect estimates provided? **6.** For the analyses in this paper, was the exposure(s) of interest measured prior to the outcome(s) being measured? **7.** Was the timeframe sufficient so that one could reasonably expect to see an association between exposure and outcome if it existed? **8.** For exposures that can vary in amount or level, did the study examine different levels of the exposure as related to the outcome (e.g., categories of exposure, or exposure measured as continuous variable)? **9.** Were the exposure measures (independent variables) clearly defined, valid, reliable, and implemented consistently across all study participants? **10.** Was the exposure(s) assessed more than once over time? **11.** Were the outcome measures (dependent variables) clearly defined, valid, reliable, and implemented consistently across all study participants? **12.** Were the outcome assessors blinded to the exposure status of participants? **13.** Was loss to follow-up after baseline 20% or less? **14.** Were key potential confounding variables measured and adjusted statistically for their impact on the relationship between exposure(s) and outcome(s)? 
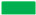
 Yes 

 No, **CD**, cannot determine; **NA**, not applicable; **NR**, not reported.

**Table 2 diagnostics-11-01626-t002:** Characteristics of patients at time of diagnosis of GC bone metastasis in the included studies.

Ref.	Country and Time Horizon Covered	Type of Study	Number of Patients	Bone Metastases Occurrence	Median/Mean Age (Years)	Menopausal Status	Primary Tumor Histopathology	Primary Tumor FIGO Stage and Grade	Survival/Follow-Up after Bone Metastasis (Months)
Ovarian Cancer									
Deng 2018 [18]	China 2010–2014	Retrospective	1481	32 patients (2.1%)	NR	NR	Serous and non-serous ovarian carcinoma	Stage IV	Median OS: 11 months
Sehouli 2013 [19]	Berlin 1994–2009	Retrospective	1717	26 patients (1.5%)	Median age: 54.0	Pre-m (*n* = 11) Post-m (*n* = 15)	Serous (*n* = 14) and non-serous (*n* = 12) ovarian carcinoma	Stage I/II (*n* = 4) Stage III/IV (*n* = 22) Grade I/II (*n* = 9) Grade III (*n* = 12) Unknown grade (*n* = 5)	Median OS: 7.2 months
Zhang 2013 [20]	China 2002–2008	Retrospective	2189	26 patients (1.1%)	Mean age: 46.7	NR	Serous ovarian carcinoma (*n* = 13) Clear cell ovarian carcinoma (*n* = 6) Ovarian germ cell carcinoma (*n* = 7)	Stage II (*n* = 3) Stage III/IV (*n* = 23)	Mean survival time: -No treatment: 3.0 months -Disodium pamidronate: 7.0 months -Chemotherapy: 8.4 months -Radionuclide therapy: 11.0 months -Chemotherapy + radiotherapy: 14.2 months -Chemotherapy + radiotherapy +disodium pamidronate: 17.3 months -Chemotherapy + radiotherapy + radionuclide therapy: 21.5 months
Zhang 2019 [21]	China 2010–2015	Retrospective	32,178	352 (1.0%)	Mean age: 65.61 ± 15.12	NR	Serous ovarian carcinoma (*n* = 87) Non-serous ovarian carcinoma (*n* = 205) Unknown ovarian carcinoma (*n* = 60)	Stage I (*n* = 23) Stage II (*n* = 46) Stage III (*n* = 129) Unknown stage (*n* = 154) Grade Ⅰ (*n* = 4) Grade Ⅱ (*n* = 7) Grade Ⅲ (*n* = 66) Undifferentiated (*n* = 34) Unknown grade (*n* = 241)	Median OS: 5 months
Endometrial Cancer									
Guo 2020 [22]	China 2010–2015	Retrospective	730	113 (15.4%)	Mean age 64.81 ± 10.92	NR	Endometrioid carcinoma Endometrial serous carcinoma Endometrial clear cell carcinoma	Stage IVB	Median OS: 15 months
Hong 2021 [23]	USA 2013–2016	Retrospective	403	18 (4.4%)	Mean age: 66.1	Post-m	Endometrial serous and non-serous carcinoma	Stage IA to IV	Median OS: 16 months
Kimyon 2016 [24]	Turkey 1993–2013	Retrospective	1345	10 (0.7%)	Mean age: 61	Post-m	Endometrioid carcinoma Endometrial serous carcinoma Endometrial clear cell carcinoma Endometrial undifferentiated carcinoma	Stage IB (*n* = 5) Stage IIIC1 (*n* = 1) Stage IIIC2 (*n* = 4) Grade I (*n* = 3) Grade II (*n* = 7) Grade III (*n* = 3)	Median OS: 17.5 months in patients with Stage IB and 4 months in Stage IIIC1–2
Li 2020 [25]	China 2010–2015	Retrospective	929	36 (3.8%)	<63 ≥63	NR	Endometrioid and non-endometrioid carcinoma	Stage IVB	Median OS: 18 months
Liu 2020 [26]	China 2010–2014	Retrospective	2948	101 (3.4%)	From 47.5 to 52.5	NR	Endometrioid and non-endometrioid carcinoma	Stage IV Grade I, II, III	Median OS: 6 months
Mao 2020 [27]	China 2010–2015	Retrospective	69,027	388 (*n* = 0.5%)	<60 (*n* = 117) 60–80 (*n* = 235) <80 (*n* = 36)	NR	Endometrioid carcinoma (*n* = 261) Endometrial serous (*n* = 46) carcinoma Endometrial carcinosarcoma (*n* = 37) Endometrial clear cell carcinoma (*n* = 17) Endometrial mixed epithelial carcinoma (*n* = 27)	Grade I (*n* = 41) Grade II (*n* = 151) Grade III (*n* = 61) Undifferentiated (*n* = 117)	Median OS: 8 months
McEachron 2020 [28]	USA 2012–2019	Retrospective	1085	10 (0.9%)	Mean age: 65	NR	Endometrioid carcinoma (*n* = 7) Endometrial serous carcinoma (*n* = 3)	Stage I (*n* = 2) Stage III/IV (*n* = 8)	Median OS: 11 months
Ouldamer 2019 [29]	France 2001–2013	Retrospective	1444	9 (0.6%)	Mean age: 75.5	Post-m	Endometrial carcinoma	Grade I (*n* = 1) Grade II (*n* = 3) Grade III (*n* = 6)	Median OS: 15 months
Takeshita 2016 [30]	Japan 1995–2014	Retrospective	926	28 (3.0%)	Mean age: 60.5	Post-m	Endometrioid and non-endometrioid carcinomaEndometrial carcinosarcoma	Stage I (*n* = 3) Stage II (*n* = 2) Stage III (*n* = 9) Stage IV (*n* = 14) Grade I/II (*n* = 9) Grade III/Others (*n* = 19)	Median OS: 6.2 months
Uccella 2013 [31]	USA 1984–2001	Retrospective	1632	19 (1.1%)	Mean age: 65	Post-m	Endometrioid carcinoma (*n* = 13) Non-endometrioid carcinoma (*n* = 6)	Stage I (*n* = 10) Stage II (*n* = 1) Stage III (*n* = 3) Stage IV (*n* = 5)	Median OS: 12 months
Yoon 2014 [32]	South Korea 1994–2012	Retrospective	1185	21 (1.7%)	Mean age: 59	NR	Endometrioid carcinoma (*n* = 8) Non-endometrioid carcinoma (*n* = 13)	Stages II/I (*n* = 4) Stages III/IV (*n* = 17) Grade I (*n* = 1) Grade II (*n* = 4) Grade III (*n* = 10) Unknown (*n* = 6)	Median OS: 15 months
Uterine Sarcoma									
Bartosh 2017 [33]	Portugal	Retrospective	130	18 (13.8%)	Mean age: 52.1 ± 9.8	NR	Uterine leiomyosarcoma	NR	Median OS: 22 months
Tirumani 2014 [34]	USA 2000–2012	Retrospective	113	37 (32.7%)	Mean age: 53	NR	Uterine leiomyosarcoma	Stage I to IV	Median OS: 45 months
Vulvar Cancer									
Prieske 2016 [35]	Germany 1996–2013	Retrospective	391	5 (1.2%)	Median age: 60	Pre-m Post-m	Vulvar squamous-cell carcinoma	Stage IA/IB Grade II/III	Median OS: 36 months
Cervical Cancer									
Kanayama 2015 [36]	Japan 1996–2010	Retrospective	713	37 (5.1%)	Median age: 58	Post-m	Cervical squamous-cell carcinoma (*n* = 30) Cervical non-squamous-cell carcinoma (*n* = 7)	Stage I/II (*n* = 18) Stage III/IV (*n* = 19)	Median OS: 12 months
Kocaer et al. 2018 [37]	Istanbul 1992–2015	Retrospective	844	18 (2.1%)	Mean age: 55.8 ± 10.0	Post-m (*n* = 14), Pre-m (*n* = 4)	Cervical squamous-cell carcinoma	Stage I/II (*n* = 6) Stage III/IV (*n* = 12) Grade I (*n* = 1) Grade II (*n* = 6) Grade III (*n* = 11)	Mean survival: 14.1 ± 7.8 months
Lin 2019 [38]	USA 1997–2017	Retrospective	1158	12 (1.0%)	Median age: 53	NR	Cervical cancer	Stage I/II Stage III/IV	Median OS: 7.0 months
Makino 2016 [39]	Japan 2000–2010	Retrospective	NR	75	Mean age: 52.2	NR	Cervical cancer	Stage I/II (*n* = 40) Stage III/IV (*n* = 34) Unknown (*n* = 1)	Median OS: 14 months in patients with bone metastases only and 5 months in patients with extra-osseous metastases
Manders 2018 [40]	Texas 2007–2014	Retrospective	349	13 (3.7%)	Median age: 55.7	Post-m Pre-m	Cervical small cell carcinoma (*n* = 1) Cervical squamous cell carcinoma (*n* = 12)	Stage III (*n* = 8) Stage IV (*n* = 5)	Median OS: 8.5 months
Matsumiya 2016 [41]	Japan 1995–2014	Retrospective	925	54 (5.8%)	Median age: 55.5	Post-m Pre-m	Cervical squamous (*n* = 42) and non-squamous cell carcinoma	Stage I/II (*n* = 21) Stage III/IV (*n* = 33)	Median OS: 22 weeks
Nartthanarung 2014 [42]	Thailand 1998–2010	Retrospective	NR	68	<45 (*n* = 13)>45 (*n* = 39)	Post-m Pre-m	Cervical carcinoma	Stage I/II (*n* = 28) Stage III/IV (*n* = 24)	Median OS: <45 years: 21 months >45 years: 34 months
Sethi 2019 [43]	India 2016	Retrospective	100	11 (11%)	NR	NR	Cervical carcinoma	NR	NR
Yin 2019 [44]	China 2006–2016	Retrospective	99	24 (24.2%)	Median age: 53	Post-m Pre-m	Cervical carcinoma	NR	Median OS: 11.7 months
Yoon 2013 [45]	Korea 1994–2011	Retrospective	2013	105 (14 excluded because of unavailable medical records) (5.2%)	Mean age: 52.5	Post-m Pre-m	Cervical carcinoma	Stage I/II (*n* = 37) Stage III/IV (*n* = 38)	Median OS: 10 months
Zhang 2018 [46]	USA 2010–2015	Retrospective	19,363	469 (2.4%)	Mean age: 56.43 ± 13.78	Post-m Pre-m	Cervical carcinoma	Stage I/II (*n* = 145) Stage III/IV (*n* = 242) Unknown (*n* = 82) Grade I (*n* = 9)Grade II (*n* = 78) Grade III (*n* = 211) Unknown (*n* = 171)	Median OS: 6 months
Zhang 2020 [47]	China 2010–2016	Retrospective	1448	520 (35.9%)	≤40: *n* = 72 41–64: *n*= 307 ≥65 years: 141	Post-m Pre-m	Cervical carcinoma	Stage IVB Grade I/II (*n* = 94) Grade III (*n* = 241) Unknown (*n* = 185)	Median OS: 10 months
Zhou 2020 [48]	China 2010–2016	Retrospective	1347	225 (16.7%)	Mean age: 57.00 ± 14.29	Post-m Pre-m	Cervical carcinoma	AJCC T Stage: T1 (*n* = 24) T2 (*n* = 39) T3 (*n* = 100) T4 (*n* = 27) TX (*n* = 35)	Median OS: 8 months

**Table 3 diagnostics-11-01626-t003:** Diagnosis and main characteristics of bone metastases in patients with GCs.

Ref.	Bone Metastases Diagnosis Method	Bone Metastases-Free Interval	Bone Metastases Site(s)	Bone Metastases Number	Bone Metastases Symptoms	Extraosseous Metastases
Ovarian Cancer						
Deng 2018 [18]	NR	NR	NR	NR	NR	23/32
Sehouli 2013 [19]	X-ray (*n* = 16), CT (*n* = 24) MRI (*n* = 16), BS (*n* = 20)	12 months (*n* = 8) >12 months (*n* = 15) At time of primary diagnosis (*n* = 3)	Vertebrae, pelvic bone, ribs, bones of the extremities and the skull	Multiple: 21 Single: 5	Pain	NR
Zhang 2013 [20]	X-ray, CT, MRI	Stage I 3 years Stage II 2 years Stage III 1 to 5 years Stage IV: after diagnosis of ovarian cancer and after 1 year	12 cervical vertebrae, 10 lumbar vertebrae, 8 pelvis, 7 thoracic vertebrae, 5 limbs, 1 ribs, 2 sternum	NR	Low back pain, thoracodynia, difficulty in walking	9 lungs, 5 liver, 4 brain, 3 splenic, 2 adrenals, 12 lymphatics
Zhang 2019 [21]	NR	NR	NR	NR	NR	Lung, liver, brain
Endometrial Cancer						
Guo 2020 [22]	NR	NR	NR	Single and multiple	NR	Yes
Hong 2021 [23]	PET/CT	6/18 months	Extremity: 1/18 Axial: 17/18	Single and multiple	NR	NR
Kimyon 2016 [24]	X-ray, CT	13 months	Costa, pelvis, sternum, tibia, scapula, skull	Single (*n* = 6) Multiple (*n* = 4)	Pain	*n* = 2
Li 2020 [25]	CT, X-ray	NR	NR	Single and multiple	NR	Yes
Liu 2020 [26]	NR	NR	NR	Single and multiple	NR	Yes
Mao 2020 [27]	NR	NR	NR	NR	NR	Yes
McEachron 2020 [28]	X-ray	2 patients at diagnosis 8 patients 14.4 months	Vertebrae, hip	Multiple: 7 Single: 3	NR	Yes
Ouldamer 2019 [29]	MRI, CT, 18-FDG PET CT	19 months	NR	NR	NR	Yes
Takeshita 2016 [30]	MRI, CT, 18-FDG PET CT	>12 months: 12 <12 months: 16	Pelvis, vertebrae, rib, clavicle, scapula, sternum, skull, tibia, femur	Yes (*n* = 15)	NR	Yes (*n* = 24)
Uccella 2013 [31]	NR	At diagnosis (*n* = 3) 19.5 months (*n* = 16)	Vertebrae (44.8%), hip (13.8%), skull, clavicle, sternum, humerus, ribs, femur, leg, calcaneus	Single (*n* = 13) and multiple (*n* = 6)	Pain, inflammation	Yes (*n* = 9)
Yoon 2014 [32]	PET CT	At diagnosis (*n* = 4) 9 months (*n* = 17)	Vertebrae, pelvis, rib, femur, acetabulum, clavicle, parietal bone, scapula, humerus	Single (*n* = 10) and multiple (*n* = 11)	Pain	Yes (*n* = 13)
Uterine Sarcoma						
Bartosh 2017 [33]	NR	36 months	NR	Single *n* = 7	NR	Yes (*n* = 11)
Tirumani 2014 [34]	CT	7 months	NR	Single and multiple	NR	Yes
Vulvar Cancer						
Prieske 2016 [35]	X-ray, CT	13 months	Vertebrae, pelvis, thigh	Single and Multiple	NR	pulmonal and hepatic (*n* = 3)
Cervical Cancer						
Kanayama 2015 [36]	X-ray, CT, MRI, FDG-PET/CT, BS	10 months	Pelvis, skull, vertebrae, rib, upper and lower extremities	Single (*n* = 25) Multiple (*n* = 12)	Pain	Yes (*n* = 29)
Kocaer et al. 2018 [37]	CT, MRI, BS	NR	Vertebrae	Single (*n* = 7) Multiple (*n* = 11)	NR	Yes (*n* = 8)
Lin 2018 [38]	FDG-PET	NR	NR	NR	NR	NR
Makino 2016 [39]	NR	At diagnosis (*n* = 15)	Vertebrae, pelvis	Single (*n* = 54) Multiple (*n* = 18)	Pain	Yes (*n* = 43)
Manders 2018 [40]	CT, PET/CT, FDG-PET, MRI	At diagnosis	Pelvis	Single (*n* = 8) Multiple (*n* = 4)	Pain	Yes (*n* = 9)
Matsumiya 2016 [41]	CT, PET/CT, FDG-PET, MRI	11.5 months	Vertebrae, pelvis	Single (*n* = 21) Multiple (*n* = 33)	NR	Yes (*n* = 50)
Nartthanarung 2014 [42]	CT, MRI, X-ray, BS	<45 years: 16 months >45 years: 26 months	Pelvis	Single (*n* = 7) Multiple (*n* = 35)	Pain	Yes (*n* = 12)
Sethi 2019 [43]	CT	NR	NR	NR	NR	Yes
Yin 2019 [44]	MRI, CT, PET-CT	NR	NR	NR	NR	Yes (*n* = 5)
Yoon 2013 [45]	MRI, CT, PET-CT	27 months	Vertebrae, mandible, tibia	Single (*n* = 45) Multiple (*n* = 46)	Pain	Yes
Zhang 2018 [46]	NR	At diagnosis (*n* = 364)	NR	NR	NR	Yes
Zhang 2020 [47]	NR	NR	NR	NR	NR	Yes
Zhou 2020 [48]	NR	NR	NR	NR	NR	Yes

Abbreviations: NR: not reported; X-ray: X-radiography, CT: computed tomography; MRI: magnetic resonance imaging; BS: bone scintigraphy; 18FDG PET-CT: Positron emission tomography with 2-deoxy-2-[fluorine-18]fluoro- D-glucose integrated with computed tomography; PET-CT: Positron emission tomography integrated with computed tomography.

**Table 4 diagnostics-11-01626-t004:** Development of complications and skeletal-related events in patients with GC bone metastases included in the review.

Ref.	SREs %	Bone Pain	Pathological Fractures	Spinal Cord Compression	Hypercalcemia	First-Line Therapy for Bone Metastases
Ovarian Cancer						
Deng 2018 [18]	NR	NR	NR	NR	NR	ChT Surgery
Sehouli 2013 [19]	NA	Yes	No	No	NR	ChT, RT
Zhang 2013 [20]	33%	*n* = 16	*n* = 8	*n* = 4	No	ChT + RT + surgery (*n* = 24) Surgery (*n* = 1)
Zhang 2019 [21]	NR	*n* = 8	Yes	No	NR	Surgery: 3 Stage II Cytoreductive surgery: 13 cases Stage III and in 10 cases Stage IV
Endometrial Cancer						
Guo 2020 [22]	NR	NR	NR	NR	NR	None
Hong 2021 [23]	NR	NR	NR	NR	NR	NR
Kimyon 2016 [24]	NR	Yes	NR	NR	NR	ChT, RT, surgery (*n* = 1)
Li 2020 [25]	NR	NR	NR	NR	NR	ChT, RT
Liu 2020 [26]	NR	NR	NR	NR	NR	Surgery
Mao 2020 [27]	NR	NR	NR	NR	NR	NR
McEachron 2020 [28]	1%	NR	*n* = 1	NR	NR	ChT, RT,
Ouldamer 2019 [29]	NR	NR	NR	NR	NR	RT
Takeshita 2016 [30]	NR	NR	No	NR	NR	ChT, RT, surgery
Uccella 2013 [31]	NR	Yes	*n* = 1	NR	NR	ChT, RT, surgery
Yoon 2014 [32]	NR	*n* = 17	*n* = 2	NR	None	ChT, RT, surgery
Uterine Sarcoma						
Bartosh 2017 [33]	NR	NR	NR	NR	NR	ChT, RT, surgery
Tirumani 2014 [34]	NR	No	NR	NR	NR	ChT, RT, surgery
Vulvar Cancer						
Prieske 2016 [35]	NR	NR	NR	NR	NR	ChT, RT, surgery
Cervical Cancer						
Kanayama 2015 [36]	NR	*n* = 15	NR	NR	NR	ChT, RT
Kocaer et al. 2018 [37]	NR	NR	NR	NR	NR	ChT, RT, surgery
Lin 2018 [38]	NR	NR	NR	NR	NR	NR
Makino 2015 [39]	NR	NR	NR	NR	NR	ChT, RT, other palliative treatment
Manders 2018 [40]	No	Yes	No (3 after radiation)	No	*n* = 3	ChT, RT, other palliative treatment
Matsumiya 2016 [41]	No	NR	NR	NR	NR	ChT, RT, surgery
Nartthanarung 2014 [42]	NR	Yes	NR	NR	NR	ChT, RT
Sethi 2019 [43]	NR	NR	NR	NR	NR	NR
Yin 2019 [44]	NR	NR	NR	NR	NR	ChT, RT, other palliative treatment
Yoon 2013 [45]	NR	Yes	NR	NR	*n* = 12	ChT, RT
Zhang 2018 [46]	NR	NR	NR	NR	NR	NR
Zhang 2020 [47]	NR	NR	NR	NR	NR	Surgery
Zhou 2020 [48]	NR	NR	NR	NR	NR	ChT, RT, surgery

Abbreviations: NR: not reported; NA: not applicable; ChT: chemotherapy; RT: radiotherapy.

**Table 5 diagnostics-11-01626-t005:** Search terms used for PubMed, Scopus, Web of Science Core Collection, and ClinicalTrials.gov.

Database	Search Term	Free Vocabulary and/or Medical Subject Headings (MeSH) Terms
PubMed	“bone metastasis” AND “ovarian cancer”	(“bone and bones”[MeSH Terms] OR (“bone”[All Fields] AND “bones”[All Fields]) OR “bone and bones”[All Fields] OR “bone”[All Fields]) AND (“metastasi”[All Fields] OR “neoplasm metastasis”[MeSH Terms] OR (“neoplasm”[All Fields] AND “metastasis”[All Fields]) OR “neoplasm metastasis”[All Fields] OR “metastasis”[All Fields]) AND (“ovarian neoplasms”[MeSH Terms] OR (“ovarian”[All Fields] AND “neoplasms”[All Fields]) OR “ovarian neoplasms”[All Fields] OR (“ovarian”[All Fields] AND “cancer”[All Fields]) OR “ovarian cancer”[All Fields]) AND (“2011/05/24”[PDAT]: “2021/05/24”[PDAT])
	“bone metastasis” AND “endometrial cancer”	(“bone and bones”[MeSH Terms] OR (“bone”[All Fields] AND “bones”[All Fields]) OR “bone and bones”[All Fields] OR “bone”[All Fields]) AND (“metastasi”[All Fields] OR “neoplasm metastasis”[MeSH Terms] OR (“neoplasm”[All Fields] AND “metastasis”[All Fields]) OR “neoplasm metastasis”[All Fields] OR “metastasis”[All Fields]) AND (“endometrial neoplasms”[MeSH Terms] OR (“endometrial”[All Fields] AND “neoplasms”[All Fields]) OR “endometrial neoplasms”[All Fields] OR (“endometrial”[All Fields] AND “cancer”[All Fields]) OR “endometrial cancer”[All Fields]) AND (“2011/05/24”[PDAT]: “2021/05/24”[PDAT])
	“bone metastasis” AND “uterine sarcoma”	(“bone and bones”[MeSH Terms] OR (“bone”[All Fields] AND “bones”[All Fields]) OR “bone and bones”[All Fields] OR “bone”[All Fields]) AND (“metastasi”[All Fields] OR “neoplasm metastasis”[MeSH Terms] OR (“neoplasm”[All Fields] AND “metastasis”[All Fields]) OR “neoplasm metastasis”[All Fields] OR “metastasis”[All Fields]) AND ((“uterin”[All Fields] OR “uterines”[All Fields] OR “uterus”[MeSH Terms] OR “uterus”[All Fields] OR “uterine”[All Fields]) AND (“sarcoma”[MeSH Terms] OR “sarcoma”[All Fields] OR “sarcomas”[All Fields] OR “sarcoma s”[All Fields])) AND (“2011/05/24”[PDAT]: “2021/05/24”[PDAT])
	“bone metastasis” AND “vulvar cancer”	(“bone and bones”[MeSH Terms] OR (“bone”[All Fields] AND “bones”[All Fields]) OR “bone and bones”[All Fields] OR “bone”[All Fields]) AND (“metastasi”[All Fields] OR “neoplasm metastasis”[MeSH Terms] OR (“neoplasm”[All Fields] AND “metastasis”[All Fields]) OR “neoplasm metastasis”[All Fields] OR “metastasis”[All Fields]) AND (“vulvar neoplasms”[MeSH Terms] OR (“vulvar”[All Fields] AND “neoplasms”[All Fields]) OR “vulvar neoplasms”[All Fields] OR (“vulvar”[All Fields] AND “cancer”[All Fields]) OR “vulvar cancer”[All Fields]) AND (“2011/05/24”[PDAT]: “2021/05/24”[PDAT])
	“bone metastasis” AND “vaginal cancer”	(“bone and bones”[MeSH Terms] OR (“bone”[All Fields] AND “bones”[All Fields]) OR “bone and bones”[All Fields] OR “bone”[All Fields]) AND (“metastasi”[All Fields] OR “neoplasm metastasis”[MeSH Terms] OR (“neoplasm”[All Fields] AND “metastasis”[All Fields]) OR “neoplasm metastasis”[All Fields] OR “metastasis”[All Fields]) AND (“vaginal neoplasms”[MeSH Terms] OR (“vaginal”[All Fields] AND “neoplasms”[All Fields]) OR “vaginal neoplasms”[All Fields] OR (“vaginal”[All Fields] AND “cancer”[All Fields]) OR “vaginal cancer”[All Fields]) AND (“2011/05/24”[PDAT]: “2021/05/24”[PDAT])
	“bone metastasis” AND “cervical cancer”	(“bone and bones”[MeSH Terms] OR (“bone”[All Fields] AND “bones”[All Fields]) OR “bone and bones”[All Fields] OR “bone”[All Fields]) AND (“metastasi”[All Fields] OR “neoplasm metastasis”[MeSH Terms] OR (“neoplasm”[All Fields] AND “metastasis”[All Fields]) OR “neoplasm metastasis”[All Fields] OR “metastasis”[All Fields]) AND (“uterine cervical neoplasms”[MeSH Terms] OR (“uterine”[All Fields] AND “cervical”[All Fields] AND “neoplasms”[All Fields]) OR “uterine cervical neoplasms”[All Fields] OR (“cervical”[All Fields] AND “cancer”[All Fields]) OR “cervical cancer”[All Fields]) AND (“2011/05/24”[PDAT]: “2021/05/24”[PDAT])
Web of Science Core Collection	bone metastasis” AND “ovarian cancer”	(TS = bone metastasis OR TS = bone neoplasm metastasis) AND (TS = ovarian cancer OR TS = ovarian neoplasms)—with Publication Year from 2011 to 2021
	“bone metastasis” AND “endometrial cancer”	(TS = bone metastasis OR TS = bone neoplasm metastasis) AND (TS = endometrial cancer OR TS = endometrial neoplasms)—with Publication Year from 2011 to 2021
	“bone metastasis” AND “uterine sarcoma”	(TS = bone metastasis OR TS = bone neoplasm metastasis) AND (TS = uterine sarcoma OR TS = uterin sarcoma)—with Publication Year from 2011 to 2021
	“bone metastasis” AND “vulvar cancer”	(TS = bone metastasis OR TS = bone neoplasm metastasis) AND (TS = vulvar cancer OR TS = vulvar neoplasms)—with Publication Year from 2011 to 2021
	“bone metastasis” AND “vaginal cancer”	(TS = bone metastasis OR TS = bone neoplasm metastasis) AND (TS = vaginal cancer OR TS = vaginal neoplasms)—with Publication Year from 2011 to 2021
	“bone metastasis” AND “cervical cancer”	(TS = bone metastasis OR TS = bone neoplasm metastasis) AND (TS = cervical cancer OR TS = cervical neoplasms)—with Publication Year from 2011 to 2021
Scopus	bone metastasis” AND “ovarian cancer”	(TITLE-ABS-KEY (bone AND metastasis) OR TITLE-ABS-KEY (bone AND neoplasm AND metastasis) AND TITLE-ABS-KEY (ovarian AND cancer) OR TITLE-ABS-KEY (ovarian AND neoplasms)) AND PUBYEAR > 2010
	“bone metastasis” AND “endometrial cancer”	(TITLE-ABS-KEY (bone AND metastasis) OR TITLE-ABS-KEY (bone AND neoplasm AND metastasis) AND TITLE-ABS-KEY (endometrial AND cancer) OR TITLE-ABS-KEY (endometrial AND neoplasms)) AND PUBYEAR > 2010
	“bone metastasis” AND “uterine sarcoma”	(TITLE-ABS-KEY (bone AND metastasis) OR TITLE-ABS-KEY (bone AND neoplasm AND metastasis) AND TITLE-ABS-KEY (uterine AND sarcoma) OR TITLE-ABS-KEY (uterin AND sarcoma)) AND PUBYEAR > 2010
	“bone metastasis” AND “vulvar cancer”	(TITLE-ABS-KEY (bone AND metastasis) OR TITLE-ABS-KEY (bone AND neoplasm AND metastasis) AND TITLE-ABS-KEY (vulvar AND cancer) OR TITLE-ABS-KEY (vulvar AND neoplasms)) AND PUBYEAR > 2010
	“bone metastasis” AND “vaginal cancer”	(TITLE-ABS-KEY (bone AND metastasis) OR TITLE-ABS-KEY (bone AND neoplasm AND metastasis) AND TITLE-ABS-KEY (vaginal AND cancer) OR TITLE-ABS-KEY (vaginal AND neoplasms)) AND PUBYEAR > 2010
	“bone metastasis” AND “cervical cancer”	(TITLE-ABS-KEY (bone AND metastasis) OR TITLE-ABS-KEY (bone AND neoplasm AND metastasis) AND TITLE-ABS-KEY (cervical AND cancer) OR TITLE-ABS-KEY (cervical AND neoplasms)) AND PUBYEAR > 2010
Cochrane Central Register of Controlled Trials	bone metastasis” AND “ovarian cancer”	ovarian cancer in Title Abstract Keyword AND bone metastasis in Title Abstract Keyword—with Cochrane Library publication date Between May 2011 and May 2021 (Word variations have been searched)
	“bone metastasis” AND “endometrial cancer”	endometrial cancer in Title Abstract Keyword AND bone metastases in Title Abstract Keyword—with Cochrane Library publication date Between May 2011 and May 2021 (Word variations have been searched)
	“bone metastasis” AND “uterine sarcoma”	uterine sarcoma in Title Abstract Keyword AND bone metastases in Title Abstract Keyword—with Cochrane Library publication date Between May 2011 and May 2021 (Word variations have been searched)
	“bone metastasis” AND “vulvar cancer”	vulvar cancer in Title Abstract Keyword AND bone metastases in Title Abstract Keyword—with Cochrane Library publication date Between May 2011 and May 2021 (Word variations have been searched)
	“bone metastasis” AND “vaginal cancer”	vaginal cancer in Title Abstract Keyword AND bone metastases in Title Abstract Keyword—with Cochrane Library publication date Between May 2011 and May 2021 (Word variations have been searched)
	“bone metastasis” AND “cervical cancer”	cervical cancer in Title Abstract Keyword AND bone metastases in Title Abstract Keyword—with Cochrane Library publication date Between May 2011 and May 2021 (Word variations have been searched)

## Data Availability

Not applicable.

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
