# Peer review of "Clinical Characteristics, Treatment Modalities, and Potential Contributing and Prognostic Factors in Patients with Bone Metastases from Gynecological Cancers: A Systematic Review"

_diagnostics, 2021, doi:10.3390/diagnostics11091626_

Round 1

Reviewer 1 Report

Thank you for the opportunity to review this manuscript.

The work by Salamanna at al concerns a review including 31 retrospective papers were included in this review for a total of 2880 patients with GCs bone metastases, 2 case-series studies, and 49 case reports on the Clinical characteristics, treatment modalities, potential contributing and prognostic factors in patients with bone metastases from gynecological cancers.

I have a few minor comments on the proposed manuscript, firstly I believe in my opinion as gynecogy oncologist that you cannot write about bone metastases and include them as FIGO III and IV because bone (distant) metastases in gynecological tumors cause the change of FIGO classification to FIGO IVB  (M1). For the credibility of the manuscript, I believe that self-citation is unnecessarily included. I wonder if such a large number of case reports does not adversely affect the quality of this review and whether it is a fatal error in the case of systematic review. However, I evaluate the idea of ​​the work and its execution positively. 

Author Response

-I have a few minor comments on the proposed manuscript, firstly I believe in my opinion as gynecogy oncologist that you cannot write about bone metastases and include them as FIGO III and IV because bone (distant) metastases in gynecological tumors cause the change of FIGO classification to FIGO IVB (M1).

As reported in the Paragraph 2.3. ‘Characteristics of the studies’ (Page 5, Lines: 134-135) and in Table 3 (Pages: 7-11) the gynecological cancer FIGO stage was related to the primary gynecological tumor diagnosis and no to bone metastases. 

-For the credibility of the manuscript, I believe that self-citation is unnecessarily included.

As suggested, we eliminated the self-citation from the manuscript.

-I wonder if such a large number of case reports does not adversely affect the quality of this review and whether it is a fatal error in the case of systematic review. However, I evaluate the idea of ​​the work and its execution positively.

We eliminated all case-reports from the manuscript.  

Reviewer 2 Report

‘Clinical characteristics, treatment modalities, potential contributing and prognostic factors in patients with bone metastases from gynecological cancers: a systematic review’

This study is interesting and valuable since it has reviewed the clinical characteristics, treatment modalities, potential contributing or prognostic factors of bone metastases from gynecological cancers (GCs), including endometrial cancer (EC), cervical cancer (CC), ovarian cancer (OC), uterine sarcoma (US), vulvar cancer (VuC), and vaginal cancer (VaC) (with the FIGO Stage III and IV). Based on systematic literature searches, the main bone metastatic lesions involved the vertebral column, pelvis, and low extremities. The median survival period (after diagnosis of bone metastases) was 3.0 - 45 months. The most common treatments included palliative care (e.g., RT and CHT, followed by surgery). The results of this review will provide some insights that should enable clinicians to render a more “personalized” and comprehensive care to this patient population.  

Some minor suggestions are enclosed:

P # 18

In this review, the main used treatment modality for GCs bone metastases are chemotherapy and radiotherapy.

Most frequently patients are treated for pain, which is the main common factor in all GC types, but existing rare instability, bone fractures, and spinal cord compression are also present. In these cases, bisphosphonates (which are characterized by ease of administration, long duration of action, safety, and effectiveness) [PLEASE,  GIVE SOME EXEMPLARY NAMES OF MEDICATIONS FROM THIS CLASS – e.g., alendronate or zoledronic acid] or RANKL inhibitor [PLEASE,  GIVE A NAME OF MEDICATION – e.g., Denosumab, a monoclonal antibody that targets a step in the process important to the maturation of osteoclasts] were also used to reduce the incidence of fractures or spinal cord compressions, and to relieve the diffuse pain [52].

P # 20

In conclusion, [PLEASE,  CONSIDER INCLUDING A STATEMENT ABOUT POSSIBLE BENEFITS OF AN EARLY INTRODUCTION OF MULTIDISCIPLINARY PALLIATIVE CARE (E.G., NUTRITIONAL, REHABILITATION, PHARMACOLOGICAL, AND PSYCHOLOGICAL CARE] in addition to RT, CHT, and surgery.

Thank you.

Author Response

Some minor suggestions are enclosed:

-P # 18

In this review, the main used treatment modality for GCs bone metastases are chemotherapy and radiotherapy.

Most frequently patients are treated for pain, which is the main common factor in all GC types, but existing rare instability, bone fractures, and spinal cord compression are also present. In these cases, bisphosphonates (which are characterized by ease of administration, long duration of action, safety, and effectiveness) [PLEASE,  GIVE SOME EXEMPLARY NAMES OF MEDICATIONS FROM THIS CLASS – e.g., alendronate or zoledronic acid] or RANKL inhibitor [PLEASE,  GIVE A NAME OF MEDICATION – e.g., Denosumab, a monoclonal antibody that targets a step in the process important to the maturation of osteoclasts] were also used to reduce the incidence of fractures or spinal cord compressions, and to relieve the diffuse pain [52].

As suggested, we added the names of medication for bisphosphonate and RANKL inhibitor classes (Page: 17, Lines: 311-312).

-P # 20

In conclusion, [PLEASE,  CONSIDER INCLUDING A STATEMENT ABOUT POSSIBLE BENEFITS OF AN EARLY INTRODUCTION OF MULTIDISCIPLINARY PALLIATIVE CARE (E.G., NUTRITIONAL, REHABILITATION, PHARMACOLOGICAL, AND PSYCHOLOGICAL CARE] in addition to RT, CHT, and surgery.

As recommended by the reviewer we added a statement about the benefits of palliative care (Page: 17, Lines: 113-320).

This manuscript is a resubmission of an earlier submission. The following is a list of the peer review reports and author responses from that submission.

Round 1

Reviewer 1 Report

Thank you for the opportunity to review this manuscript.
The work by Salamanna at al concerns a review including 82 papers on the  clinical characteristics, treatment modalities, potential contributing and prognostic factors of bone metastases from gynecological cancers (GCs) 

I have concerns though whether this manuscript will contribute to the evidence.

  1. In my opinion the literature older then 2010 year schould not be used.
  2. Self-citation - ref. 103
  3. The use of more then half of the #case reports# is a fatal error in the systematic review  (AMSTAR scale - critically low).
  4. Bone metastases are in all gynecological cancer FIGO stage IVb (not III and IV) - unnecessary information about another stages.
  5. #The findings of this review give a first dataset for a greater understanding of GCs bone metastases that could be able to guide clinicians towards a more “personalized” patient management and allow a better management of their patients during the disease.#                      - It will not improve the result of treatment, because they are very different neoplasms with different biology, types and treatment. The only thing they have in common is that they occur in women. 

Author Response

The work by Salamanna at al concerns a review including 82 papers on the  clinical characteristics, treatment modalities, potential contributing and prognostic factors of bone metastases from gynecological cancers (GCs).  I have concerns though whether this manuscript will contribute to the evidence. In my opinion the literature older then 2010 year schould not be used.

We agree with the reviewer, in fact in our systematic review we included studies from May 24, 2011 to May 24, 2021, as reported at Page 6, Lines 429-430.

-Self-citation - ref. 103

We used this citation because, to our knowledge, this is the only papers that critically review the interaction between bone metastases and pre-existing estrogen deficiency due to osteoporosis. We published numerous papers on bone metastases, and we have avoided all self-citations. We self-cited ref. 103 because it is really useful and relevant for the manuscript discussion.

-The use of more then half of the #case reports# is a fatal error in the systematic review (AMSTAR scale - critically low).

Since case-reports and case-series have profoundly influenced the medical literature and continue to advance our knowledge in the present time, we decided to include these types of studies in our search strategy, as also done in other systematic review on gynecological cancer bone metastases (Taiwan J Obstet Gynecol. 2017 Feb;56(1):1-8; Int J Surg. 2015 Jul;19:1-5; Gynecol Oncol. 2014 Jun;133(3):632-9). However, the data and results obtained from the various case-reports were always reported in the manuscript as anecdotal evidence, thus emphasizing the type of data and how to interpret it. Concerning the AMSTAR scale it consists of 11 items, each of which is categorized into a standardized set of four possible responses: “yes,” “no,” “can’t answer,” or “not applicable.” As reported in literature the items are related to a priori design, study selection and data extraction, the literature search, gray literature, the list of included and excluded studies, study characteristics, critical appraisal, formulation of conclusions, the combination of study results, publication bias, and conflicts of interest. Applying this scale to our review the result is not ‘critically low’.

-Bone metastases are in all gynecological cancer FIGO stage IVb (not III and IV) - unnecessary information about another stages.

As reported in the Paragraph 2.3. ‘Characteristics of the studies’ and in Table 3 the gynecological cancer FIGO stage was related to the primary tumor diagnosis.

-#The findings of this review give a first dataset for a greater understanding of GCs bone metastases that could be able to guide clinicians towards a more “personalized” patient management and allow a better management of their patients during the disease.#                      - It will not improve the result of treatment, because they are very different neoplasms with different biology, types and treatment. The only thing they have in common is that they occur in women.

We agree with the review that the heterogeneity of gynecological cancer types/subtypes, their histopathology, and also the variability across geographical regions (including screening method and missing, underreported, or incorrect data) make it difficult to interpret the data. However, having specific information related to clinical dataset, patients’ specific characteristics, prognostic and predictive factors among patients’, metastatic tumor characteristics and specific treatment could help researchers and clinicians to the identification of novel and more advantageous prognostic factors that could ideally translate in better patient stratification. After the reviewer suggestion we clarified the sentence as follow: “To our knowledge, this is the first review that has been focused on clinical characteristics, treatments, potential contributing and prognostic factors of bone metastases, in different types of GCs. The findings of this systematic review give the first dataset for a greater understanding of GCs bone metastases that could be able to guide clinicians towards a more “personalized” patient management. We look forward to future prospective and large population-based research on this complex issue, which we hope will improve and in-crease the quality of life and patient survival time.” (Page 4, Lines 331-337).

Reviewer 2 Report

This article provides an interesting review of the clinical characteristics, treatment modalities, contributing, and prognostic factors in patients with bone metastases from gynecological cancers.

Feedback is provided to the authors [below; in red].

Please, consider these suggestions/modification [e.g., grammar, style, clarity, etc.].

P#1

  1. Introduction

Bone is a common site of metastases and frequently indicates a short-term prognosis in patients with cancer.

Please, use the phrase: ‘patients with cancer’ in the entire article [the phrase: cancer patients’- can be stigmatizing].

P#2 top

It is estimated that in the United States, the number of women with GCs is approximately 80,000/year 47 [3].

Despite the enormous progress and advancements in the preventive, diagnostic, and therapeutic interventions for GCs, distant metastases and recurrence are still the leading cause of death in patients. Metastases from GCs differ depending on the cancer type.

P#19

2.4. Bone metastases diagnosis and main characteristics

EC bone metastases were mostly diagnosed with CT, MRI, and X-ray, followed by PET/CT and BS.

P#29

3.Discussion

Additionally, the heterogeneity of GC types/subtypes, their histopathology, and the variability across geographical regions (including screening method and missing, underreported, or incorrect data) further complicate the understanding of certain clinical features and specific therapeutic approaches to GCs bone metastases.

P#35

  1. Conclusions

To our knowledge, this is the first review that has been focused on clinical characteristics, treatments, potential contributing and prognostic factors of bone metastases, in different types of GCs. The findings of this systematic review give the first dataset for a greater understanding of GCs bone metastases that could be able to guide clinicians towards a more “personalized” patient management and allow better management of their patients during the disease  course

We look forward to future prospective and large population-based research on this complex issue, which we hope will improve and increase the quality of life and patient survival time.

A final suggestion:

To guide clinicians towards a more “personalized” patient management and allow comprehensive care of their patients with GCs bone and metastases, the authors may consider adding a brief comment about palliative care, which should be provided by a treatment team, including a gynecologic oncologist, a radiation oncologist, a radiologist, an orthopedic surgeon, a pain specialist, and a palliative care physician. Combining the skills of these experts is very useful for optimal therapy for bone metastases that can cause severe pain, compression of the spinal column or nerve roots, bone fractures, and hypercalcemia. Moreover, diffuse bone pain (with no evidence of fracture) may be treated with bisphosphonates (which are characterized by ease of administration, long duration of action, safety, and effectiveness) or denosumab (a RANKL inhibitor). These bone-modifying agents can be used to reduce the incidence of fractures or spinal cord compressions, and also to relieve pain.

Ref.

1.Mullen MM, Divine LM, Porcelli BP, Wilkinson-Ryan I, Dans MC, Powell MA, et al. The effect of a multidisciplinary palliative care initiative on the end of life care in gynecologic oncology patients. Gynecol Oncol. 2017; 147 (2):460-464.

  1. Lopez-Acevedo M, Lowery WJ, Lowery AW, Lee PS, Havrilesky LJ. Palliative and hospice care in gynecologic cancer: A review. Gynecol Oncol. 2013;131(1):215-21.

Author Response

- This article provides an interesting review of the clinical characteristics, treatment modalities, contributing, and prognostic factors in patients with bone metastases from gynecological cancers.

Feedback is provided to the authors [below; in red].

Please, consider these suggestions/modification [e.g., grammar, style, clarity, etc.].

P#1

Introduction

Bone is a common site of metastases and frequently indicates a short-term prognosis in patients with cancer. Please, use the phrase: ‘patients with cancer’ in the entire article [the phrase: cancer patients’- can be stigmatizing].

P#2 top

It is estimated that in the United States, the number of women with GCs is approximately 80,000/year 47 [3]. Despite the enormous progress and advancements in the preventive, diagnostic, and therapeutic interventions for GCs, distant metastases and recurrence are still the leading cause of death in patients. Metastases from GCs differ depending on the cancer type.

P#19

2.4. Bone metastases diagnosis and main characteristics

EC bone metastases were mostly diagnosed with CT, MRI, and X-ray, followed by PET/CT and BS.

P#29

3.Discussion

Additionally, the heterogeneity of GC types/subtypes, their histopathology, and the variability across geographical regions (including screening method and missing, underreported, or incorrect data) further complicate the understanding of certain clinical features and specific therapeutic approaches to GCs bone metastases.

P#35

Conclusions

To our knowledge, this is the first review that has been focused on clinical characteristics, treatments, potential contributing and prognostic factors of bone metastases, in different types of GCs. The findings of this systematic review give the first dataset for a greater understanding of GCs bone metastases that could be able to guide clinicians towards a more “personalized” patient management and allow better management of their patients during the disease course. We look forward to future prospective and large population-based research on this complex issue, which we hope will improve and increase the quality of life and patient survival time.

We have performed all the suggestions/modifications proposed by the reviewer.

-A final suggestion:

To guide clinicians towards a more “personalized” patient management and allow comprehensive care of their patients with GCs bone and metastases, the authors may consider adding a brief comment about palliative care, which should be provided by a treatment team, including a gynecologic oncologist, a radiation oncologist, a radiologist, an orthopedic surgeon, a pain specialist, and a palliative care physician. Combining the skills of these experts is very useful for optimal therapy for bone metastases that can cause severe pain, compression of the spinal column or nerve roots, bone fractures, and hypercalcemia. Moreover, diffuse bone pain (with no evidence of fracture) may be treated with bisphosphonates (which are characterized by ease of administration, long duration of action, safety, and effectiveness) or denosumab (a RANKL inhibitor). These bone-modifying agents can be used to reduce the incidence of fractures or spinal cord compressions, and also to relieve pain.

Ref.

1.Mullen MM, Divine LM, Porcelli BP, Wilkinson-Ryan I, Dans MC, Powell MA, et al. The effect of a multidisciplinary palliative care initiative on the end of life care in gynecologic oncology patients. Gynecol Oncol. 2017; 147 (2):460-464.

Lopez-Acevedo M, Lowery WJ, Lowery AW, Lee PS, Havrilesky LJ. Palliative and hospice care in gynecologic cancer: A review. Gynecol Oncol. 2013;131(1):215-21.

As suggested by the reviewer we added a brief comment about palliative care and bone-modifying agents (also adding the suggested references) in the discussion section: “The main used treatment modality for GCs bone metastases is radiotherapy. Most frequently patients are treated for pain, that is the main common factor in all GC types, but existing rare instability, bone fractures and spinal cord compression are also present. In these cases, bisphosphonates (which are characterized by ease of administration, long duration of action, safety, and effectiveness) or RANKL inhibitor were also used to reduce the incidence of fractures or spinal cord compressions, and to relieve the diffuse pain [102]. In addition, to give a relief of suffering and provision of the best possible quality of life for both the patient and her family, no matter where she is in her treatment course, palliative care was also applied [103-104].” (Page 5, Lines: 363-369).